# Effect of Clinical Parameters on Risk of Death from Cancer after Radical Prostatectomy in Men with Localized and Locally Advanced Prostate Cancer

**DOI:** 10.3390/cancers14082032

**Published:** 2022-04-18

**Authors:** Daimantas Milonas, Tomas Ruzgas, Zilvinas Venclovas, Daniele Jonusaite, Aivaras Jonas Matijosaitis, Darius Trumbeckas, Edmundas Varpiotas, Stasys Auskalnis, Darijus Skaudickas, Ramunas Mickevicius, Kestutis Vaiciunas, Jonas Mickevicius, Mindaugas Jievaltas

**Affiliations:** 1Department of Urology, Medical Academy, Lithuanian University of Health Sciences, 44307 Kaunas, Lithuania; zilvinas.venclovas@stud.lsmu.lt (Z.V.); aivaras.matijosaitis@lsmuni.lt (A.J.M.); darius.trumbeckas@lsmuni.lt (D.T.); edmundas.varpiotas@gmail.com (E.V.); stasys.auskalnis@lsmuni.lt (S.A.); darijus.skaudickas@lsmuni.lt (D.S.); ramunas.mickevicius@lsmuni.lt (R.M.); kestutis.vaiciunas@lsmuni.lt (K.V.); jonas.mickevicius@gmail.com (J.M.); mindaugas.jievaltas@lsmuni.lt (M.J.); 2Department of Applied Mathematics, Kaunas University of Technology, 44249 Kaunas, Lithuania; tomas.ruzgas@ktu.lt; 3Faculty of Medicine, Medical Academy, Lithuanian University of Health Sciences, 44307 Kaunas, Lithuania; daniele.jonusaite@stud.lsmu.lt

**Keywords:** prostate cancer, clinical factors, radical prostatectomy, cancer-specific mortality

## Abstract

**Simple Summary:**

Since prostate cancer-related deaths hold the third most common cause worldwide in men, it is important to acknowledge clinical predictors leading to such high cancer-related mortality rates. The study aimed to assess these predictors and to identify patient groups with increased risk of cancer-specific mortality. This retrospective cohort study included patients who underwent radical prostatectomy due to localized and locally advanced prostate cancer. We analyzed cancer-specific and other-cause mortality rates regarding the most important factors, such as Grade Group, pathological stage, patient’s age, and various combinations of these factors. The presented long-term mortality plots can be useful in daily practice and help clinicians identify patient cohorts with the most aggressive prostate cancer, who could benefit from more intensive or novel multimodal treatment strategies.

**Abstract:**

Background: The study aimed to assess predictors and to identify patients at increased risk of prostate-cancer-specific mortality (CSM) after radical prostatectomy (RP). Methods: A total of 2421 men with localized and locally advanced PCa who underwent RP in 2001–2017 were included in the study. CSM predictors were assessed using multivariate competing risk analysis. Death from other causes was considered a competing event. Cumulative CSM and other-cause mortality (OCM) were calculated in various combinations of predictors. Results: During the median 8 years (interquartile range 4.4–11.7) follow-up, 56 (2.3%) of registered deaths were due to PCa. Cumulative 10 years CSM and OCM was 3.6% (95% CI 2.7–4.7) and 15.9% (95% CI 14.2–17.9), respectively. The strongest predictors of CSM were Grade Group 5 (GG5) (hazard ratio (HR) 19.9, *p* < 0.0001), lymph node invasion (HR 3.4, *p* = 0.001), stage pT3b-4 (HR 3.1, *p* = 0.009), and age (HR 1.1, *p* = 0.0007). In groups created regarding age, stage, and GG, cumulative 10 years CSM ranged from 0.4–84.9%, whereas OCM varied from 0–43.2%. Conclusions: CSM after RP is related to GGs, pathological stage, age, and combinations of these factors, whereas other-cause mortality is only associated with age. Created CSM and OCM plots can help clinicians identify patients with the most aggressive PCa who could benefit from more intensive or novel multimodal treatment strategies.

## 1. Introduction

In the European Union, the predicted mortality rate for prostate cancer (PCa) in 2020 was 10.0/100,000 and despite decreasing by 7% since 2015, it remains the third most common cause of death in men [1]. Identifying risk for cancer-specific mortality (CSM) and other-cause mortality (OCM) is essential for personalized clinical decision-making in PCa management.

In population-based PCa cohorts, CSM variates from 0.4 to 70% at 10 years follow-up and depends on initial clinical and pathological parameters, comorbidity, age, and received treatment [2,3,4,5]. In most cases, locally advanced PCa has a protracted natural history. According to available reports, overall 10 or 15 years CSM ranged from 2.8% [3] to 7% [6] after RP and from 6% [3] up to 29% [2,7] in untreated patients. However, in patients with different unfavorable cancer characteristics, 10 years CSM increased up to 25% [6,8,9].

Numerous pre- and postoperative prognostic models have been suggested for the prediction of disease progression and were analyzed recently [10]. The authors concluded that D’Amico’s preoperative risk nomogram [11] and Eggener’s postoperative nomogram [6] could be used as optimal externally validated tools for CSM, but these nomograms were validated only in single studies. Genomic classifiers were also proposed for use in some settings [12,13,14]. However, the implementation of these classifiers into daily clinical practice is questionable. Therefore, the identification of the most aggressive or potentially lethal PCa remains challenging.

According to the guidelines of the European Association Urology, International Society of Urological Pathology (ISUP) Grade Group (GG), stage, and preoperative prostate-specific antigen (PSA) are the most important predictors of disease progression [15]. Low stage, GG1, and PSA < 10 ng/mL are associated with low-risk PCa, whereas stage ≥ T3 and/or GG4–5 and/or PSA > 20 ng/mL represent the high-risk PCa. However, within risk groups, various combinations of predictors are possible and the range of CSM in different subsets could variate significantly [8,9,16,17]. Regarding the analysis of survival rates in men with PCa, information of the competing risk of other-cause mortality is important as well. In low-risk patients, the main cause of death is not related to PCa. On the other hand, a not-negligible proportion of men with aggressive PCa may have died from other causes. The estimated proportion between CSM and OCM in different risk groups is important in the understanding of the natural course of surgically treated PCa patients and counselling patients for further treatment.

In this study, we determined the key predictors of long-term CSM in men after RP using competing risk analysis. We then quantified long-term cumulative mortality rates in various combinations of the three main predictors. We hypothesized that the created subsets could have a high risk of CSM and a low risk of OCM and vice versa. These data could be helpful to identify candidates for the most aggressive and potentially lethal disease course and to manage additional, more intensive treatment.

## 2. Material and Methods

Between 2001 and 2017, 2421 men underwent RP for clinically localized or locally advanced PCa at a single tertiary university center. The data of all patients were registered in a PCa database. Clinical and pathological cancer characteristics were collected before and after surgery. For PCa grading, the 2014 ISUP GG model was used [18]. Prospective data collection was approved by the University ethical committee.

The death from any cause was the main endpoint of study. The day and cause of death was received from the national database and rechecked using center database. Only cases without clinically and radiologically approved progression were defined as other-cause death.

Chi-square and *t*-test were used for the analysis of categorical and quantitative variables, respectively. The death of other causes was accounted as a competing risk. The Fine and Gray model was used for competing risk regression analysis and identification of most important predictors of CSM, and accounting 5 years and 10 years cumulative incidence of CSM and OCM using pathological stage, and patient’s age at RP. Pathological stage and age were stratified to pT2-3a vs. pT3b-4 and <65 vs. ≥65 years, respectively. Lymph node dissection was performed in 33% of the study men; therefore, lymph node status was excluded from the additional analysis. Despite that SM was detected as a significant predictor of CSM in multivariate analysis, it is more a surgeon-dependent factor rather than a clinical cancer characteristic and was not included in further analysis. Cumulative incidence of CSM and OCM was quantified in different groups.

SAS software (version 9.4) was used for statistical analysis, and a two-sided significance level *p* < 0.05 was chosen as significant.

## 3. Results

Patient characteristics are presented in Table 1. At a median 91 (IQR 52–135) months of follow-up for survivors, 353 (14.6%) deaths were registered, of which 56 (2.3%) were related to PCa and 11 (0.5%) of which happened within 90 days after RP. Younger men (up to 65 years old) showed the trend to have less aggressive PCa features regarding preoperative PSA, pathological GG, and positive lymph node rate (Appendix A) compared with older counterparts. Cumulative 10 years CSM and OCM were 3.6% (95% CI 2.7–4.7) and 15.9% (95% CI 14.2–17.9), respectively. Highest 10 years CSM associated with age ≥ 65 years was 5.4% (95% CI 3.9–7.6); stage pT3b-4 was 25.1% (95% CI 18.3–34.4); positive SM was 8.4% (95% CI 6.3–11.2); lymph node invasion was 45.1% (95% CI 30.0–67.8); and GG5 was 46.4% (95% CI 32.2–66.6). Higher OCM at 10 years was associated mostly with older age (21.4%, 95% CI 18.6–25.1); meanwhile, different cancer characteristics did not influence OCM. According to various parameters, cumulative 5 years and 10 years mortality data are presented in Table 2.

Multivariate competing risk regression analysis depicted the main predictors of CSM (Table 3), the most important of which were GG4 (HR 8.2, 95% CI 1.87–24.76, *p* = 0.001) and GG5 (HR 19.9, 95% CI 5.55–74.31, *p* < 0.0001) followed by age (HR 1.1, 95% CI 1.03–11.14, *p* = 0.0007), lymph node invasion (HR 3.4, 95% CI 1.59–7.17, *p* = 0.001), pathological stage pT3b-4 (HR 3.1, 95% CI 1.32–7.38, *p* = 0.009), and SM status (HR 2.4, 95% CI 1.24–4.74, *p* = 0.009).

Various combinations were created using the most significant predictors (GGs 1 to 5, pathological stage pT2-3a vs. pT3b-4, and age <65 vs. ≥65 years) of CSM. Mortality data and plots are presented in Appendix A and Figure 1.

Cumulative 10 years CSM varies from 0.4% in younger men with favorable cancer features (GG1 and pT2-3a) to 85% in older men with the most aggressive cancer characteristics (GG5 and pT3b-4). The risk of CSM was higher in older men compared to the younger subset when other cancer characteristics were the same. The same trend was seen in all possible GGs and pT stage combinations (Appendix A). Similarly, men with stage pT3b-4 had an increased risk of CSM in comparison with pT2-3a at any combination of age and GG. In GGs 1 to 5, the risk of CSM increased gradually with worse outcomes in GG5. For men at the same age and stage, higher GGs were associated with an increased risk of CSM.

Cumulative 10 years OCM varies from 0 to 43.2% and is associated mostly with age rather than with unfavorable cancer features. Indeed, the risk of death from other causes was from 10- to 20-fold higher in men with the most favorable cancer characteristics, whereas in the combination of pT3b-4 and GG5, independent of age, all patients died from PCa (Appendix A, Figure 1). 

## 4. Discussion

Predicting CSM and OCM after RP is essential in clinical practice to avoid unnecessary secondary treatment and to identify candidates for more intensive multimodal salvage treatment. Our primary goal was to assess predictors of CSM in a large single-center cohort and quantify cumulative long-term mortality in subsets based on a combination of main predictors. The study results provided information about the natural history of surgically treated PCa and presented mortality plots that could be useful and easy to use in daily practice for patient counselling.

We observed that combinations of pathological features have a variable effect on the risk of CSM. We found that GG is a key predictor and CSM increased gradually following from GG1 to GG5. In men with pathological GG1, the combination with stage pT2-3a was found in 99.7% of the GG1 cohort with extremely low (up to 1%) risk of 10 years CSM. A similar trend was seen in the GG2 subset. The combination of GG2 and pT2-3a stage comprised 95.3% of the subset, and the risk of death from cancer at 10 years reached up to 2.5%. Such findings demonstrate that men with pathological GG1–2 have a very low potential to progress, and death from other causes is more likely than death from cancer.

Very recently, EAU guidelines suggested stratifying the intermediate-risk PCa into low-intermediate- and high-intermediate-risk subgroups because of different outcomes in GG2 and GG3 cohorts [19]. Our study demonstrates the variability of combinations and other parameters with GG2 and GG3 and the impact of these combinations on outcomes. Men with GG2 and pT2-3a and younger men with GG3 and pT2-3a had a similar (up to 3%) risk of 10 years CSM. Meanwhile, patients with GG2 and stage pT3b-4, as well as older men with GG3 and pT2-3a or younger patients with GG3 and pT3b-4, had from 5 to 12% CSM risk at 10 years. Moreover, in older men with GG3 and pT3b-4, 10 years CSM increased up to 26%. Indeed, men with such a combination of unfavorable cancer features had a similar risk of 10 years CSM as younger men with GG4 and pT3b-4 or any age patients with GG5 and pT2-3a. Such heterogeneity between CSM in GG2 and GG3 cohort, taking into account possible upstaging and upgrading after RP, demonstrates a need for a precise stratification of the intermediate-risk patients before initial treatment is recommended.

Men with high-risk PCa are potential candidates for multimodal treatment. The definition of this population used by various medical associations is based on the same clinical parameters: GG4–5, stage >T2c, and preoperative PSA >20 ng/mL [15,20,21,22]. There is a shred of evidence that high-risk PCa patients have a different risk of disease progression that depends on dominant unfavorable cancer features [8,16,23]. Despite various data demonstrating the heterogeneity of this population [24], until now there are no generally accepted criteria for how to stratify this population into subgroups. Our analysis shows that only men with GG4 and pT3b-4, and men with GG5 and any stage, had a higher risk of 10 years CSM compared to OCM and could be defined as a high-risk PCa patient cohort. Moreover, men with GG5 and stage pT3b-4 could be identified as very high-risk patients, as men in this subgroup had an extremely high (>55%) risk of 10 years CSM. Our findings are in concordance with the recommendation of NCCN, where men with stage T3b-4 are attributed to a very-high-risk subgroup [25].

PCa patient long-term survival data are available from various population-based studies [2,3,4,26], multi-institutional series [6,8,16], or large cohort of a single-center [9]. In the majority of these studies, the main predictors of CSM are similar to those detected in our study (Gleason score, stage, age, PSA) and depend on the main goal and patient characteristics included in the analysis. In general, our detected overall 10 years CSM and OCM (3.6 and 15.9%, respectively) were at the same range compared with 3.2 and 5.9% in 8741 men treated with RP at a single center [9], or 5.8 and 21.5% rates in 22,244 men presented by Abdollah et al. [3]. However, there is some difference comparing 10 years CSM according to main PCa features. Our study found a 25% risk of 10 years CSM in stage pT3b-4, 45% CSM in men with positive lymph nodes, 16% in GG4, and 46% in GG5. Meanwhile, Eggener et al. presented 8.4 to 13% CSM in stage pT3b, 12% to 23% in LNI, and 13% to 18% in Gleason score 8–10 mortality from cancer in LNI according to the different age groups of 23,910 men after RP [6]. Indeed, the majority of the studies presented outcomes comparing the combination of PCa risk factors: low-, intermediate-, and high-risk groups based on D’Amico criteria [2,4,9,26] or created new prognostic groups based on significant risk factors assessed in the study [8,16,17]. Independent of definition, patients with worse prognosis demonstrated 13 to 24% 10 years CSM [8,9,16,17]. In our study, patients with at least two of the most unfavorable PCa characteristics (GG4 and pT3b-4 or GG5 and pT3b-4) were from 20.5 to 84.9% at risk of 10 years CSM, which is higher when compared with aforementioned studies and very likely represents the most aggressive disease subgroup. One possible explanation of such differences in mortality rates could be a different interpretation of the Gleason score. In our study, we analyzed all GGs separately; meanwhile, in other studies, Gleason scores of 8–10 were joined together into one subgroup. The difference in outcomes among patients with GG4 vs. GG5 was clearly shown in some previous studies [27,28].

Taken together, the study findings presented herein demonstrate different risks of CSM and OCM in men after initial surgical treatment. The challenge nowadays is to better understand the mechanism of cancer development, find real PCa markers, the best imaging exam, and develop new specific therapies in order to reduce the mortality of PCa [29]. However, clinical cancer characteristics remain very important and mostly used predictors of CSM. A suggested combination of cancer characteristics as well as age could be helpful to identify men with the lowest and highest risk of disease progression. Moreover, our survival data show heterogeneity within generally accepted low-, intermediate-, and high-risk groups as well as the necessity of more precise stratification models. Our analysis did not show the significance of PSA for the prediction of CSM. However, a combination of other traditionally used clinical parameters, such as stage, GG, and, according to our analysis, age, is very important in patient stratification. Younger men demonstrated a trend to have less aggressive pathological cancer characteristics compared with older men in our cohort. Possibly, more intensive salvage treatment in younger men could be another explanation for age impact on CSM. On the other hand, better response to received treatment in younger men could be more important than treatment intensiveness or some cancer characteristics on CSM. The easy-to-use presented survival plots in daily clinical practice are one of the advantages of our study.

Several methodological issues limited our study. First, a comorbidity assessment was not performed. Therefore, our main analysis was focused on CSM. Indeed, men who underwent surgical treatment mostly are healthy with Charlson comorbidity index of 0 approximately in 75% [4,17] and with a minimal impact on CSM [17]. Second, we did not include patients who received other initial PCa treatment methods. Therefore, recommendations from our study cannot be directly applied to men who underwent other primary treatment. On the other hand, the ProtectT trial demonstrated a low 1% risk of 10 years CSM irrespective of the treatment group [30,31], and it is very likely that our detected mortality rates could represent outcomes after any active initial treatment. Third, the retrospective study design resulted in some missing values for patient characteristics and disease progression during the follow-up. Forth, specimen grade was not re-evaluated following the ISUP 2014 recommendations, and Gleason score was directly transformed to GG. However, in the majority of publications, GGs were created in the same way [27]. Finally, through the time of the study (20 years), significant changes in surgical approach and adjuvant or salvage treatment of disease recurrence have occurred. However, all changes in the management of the patients were performed following existing PCa treatment guidelines.

## 5. Conclusions

This study presents the stratification of PCa patients into demarcated subgroups according to GG, pathological stage, and age. The presented long-term cancer-specific mortality and other-cause mortality plots are easy to use and can help clinicians identify the patient cohort with the most aggressive PCa who could benefit from more intensive or novel multimodal treatment strategies.

## Figures and Tables

**Figure 1 cancers-14-02032-f001:**
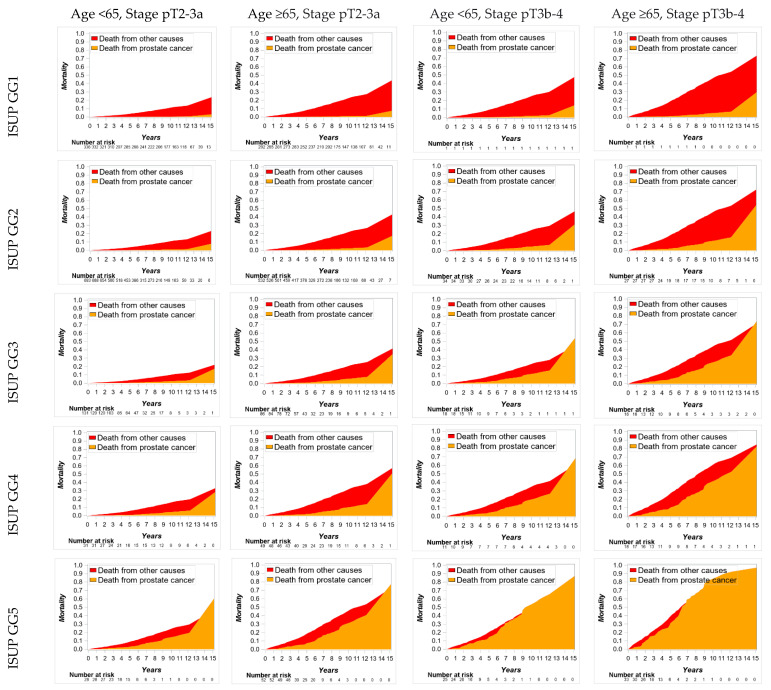
Cumulative cancer-specific and other-cause mortality in groups stratified by patient age (<65 vs. ≥65), pathological stage (pT2-3a vs. pT3b-4), and postoperative International Society of Urological Pathology Grade Group (ISUP GG).

**Table 1 cancers-14-02032-t001:** Descriptive characteristics of 2421 prostate cancer patients treated with radical prostatectomy.

Parameter	All Patients
	*n* = 2421
Age, years—median (IQR)	64 (59–68)
<65 years, *n* (%)	1309 (56)
≥65 years, *n* (%)	1112 (46)
PSA, ng/mL—median (IQR)	6.3 (4.7–9.5)
Clinical stage (cT), *n* (%)	
T1	705 (29.1)
T2	1397 (57.7)
T3	313 (12.9)
Unknown	6 (0.3)
Biopsy GG, *n* (%)	
1	1395 (57.6)
2	754 (31.1)
3	102 (4.2)
4	104 (4.3)
5	52 (2.1)
Unknown	14 (0.7)
Pathological stage (pT), *n* (%)	
T2	1574 (65.0)
T3a	663 (27.4)
T3b-4	184 (7.6)
Pathological GG, *n* (%)	
1	630 (26.0)
2	1286 (53.1)
3	251 (10.4)
4	109 (4.5)
5	138 (5.7)
Unknown	7 (0.3)
Surgical margins status, *n* (%)	
negative	1617 (66.8)
positive	698 (28.8)
Unknown	106 (4.4)
Lymph nodes status, *n* (%)	
pN0	713 (29.4)
pN1	83 (3.4)
Unknown	1625 (67.2)
Post RP treatment, *n* (%)	
Adjuvant RT	63 (2.6)
Salvage ADT	90 (3.7)
Salvage RT ± ADT	410 (16.9)

PSA—prostate-specific antigen, GG—International Society of Urological Pathology Grade Groups, RP radical prostatectomy, pN0—negative lymph node, pN1—positive lymph node, RT—radiotherapy, ADT—androgen deprivation therapy.

**Table 2 cancers-14-02032-t002:** Cumulative 5- and 10-year mortality from cancer and other causes according to age, pathological stage, surgical margin status, lymph node status, and grade group.

Parameter	5 Years Mortality (95% CI)		10 Years Mortality (95% CI)	
	Prostate Cancer	Other Causes	Prostate Cancer	Other Causes
All Study Patients	1.1 (0.8–1.6)	6.1 (5.3–7.1)	3.6 (2.7–4.7)	15.9 (14.2–17.9)
Age (years)				
<65	0.6 (0.4–1.0)	4.0 (3.1–5.2)	1.9 (1.1–3.3)	10.9 (8.7–12.9)
≥65	1.7 (1.1–2.6)	8.4 (7.0–10.4)	5.4 (3.9–7.6)	21.4 (18.6–25.1)
Pathological Stage				
pT2	0.4 (0.2–0.6)	5.6 (4.8–6.7)	1.3 (0.8–2.2)	15.6 (13.1–17.0)
pT3a	1.2 (0.7–2.2)	6.1 (4.9–7.9)	4.3 (2.6–7.2)	16.3 (12.9–20.4)
pT3b-4	7.9 (4.5–13.6)	10.0 (7.2–15.4)	25.1 (18.3–34.4)	19.6 (13.3–27.5)
Surgical Margins				
Negative	0.4 (0.3–0.7)	5.9 (5.0–7.0)	1.4 (0.9–2.1)	16.1 (13.5–17.7)
Positive	2.7 (1.7–4.3)	6.6 (5.4–8.4)	8.4 (6.3–11.2)	16.4 (14.5–20.8)
Lymph Node Invasion				
Unknown	0.4 (0.2–0.8)	5.8 (4.9–7.1)	1.6 (1.0–2.6)	16.1 (13.5–17.9)
Negative	1.2 (0.8–2.0)	6.2 (5.0–7.7)	4.5 (3.0–6.9)	16.2 (13.6–19.4)
Positive	14.9 (8.7–25.6)	9.4 (5.6–17.8)	45.1 (30.0–67.8)	11.9 (7.9–23.5)
Grade Groups (GG)				
GG1	0.1 (0.04–0.4)	5.7 (4.8–7.6)	0.7 (0.02–2.0)	16.1 (13.1–19.1)
GG2	0.4 (0.2–0.8)	5.7 (4.7–7.3)	2.1 (1.3–3.5)	15.9 (13.3–17.8)
GG3	1.4 (0.6–3.1)	5.2 (3.2–8.3)	6.7 (2.8–15.9)	12.6 (8.8–20.5)
GG4	3.4 (1.4–7.9)	9.4 (5.3–14.9)	15.6 (9.1–26.7)	19.3 (12.3–30.7)
GG5	11.8 (7.2–19.4)	9.2 (5.1–14.7)	46.4 (32.2–66.6)	5.9 (0.3–20.6)

**Table 3 cancers-14-02032-t003:** Multivariable competing risk analysis of cancer specific mortality.

10 Years Cancer-Specific Mortality			
Parameter	HR	95% CI	*p*
Preoperative PSA (ng/mL)	0.98	0.95–1.02	0.3
Age (year)	1.1	1.03–1.14	0.0007
Pathological stage			
pT2		Referent	
pT3a	1.2	0.52–2.66	0.7
pT3b-4	3.1	1.32–7.38	0.009
Pathological ISUP GG			
GG 1		Referent	
GG2	2.2	0.58–6.02	0.2
GG3	4.8	1.18–18.47	0.02
GG4	8.2	1.87–24.76	0.001
GG5	19.9	5.55–74.31	<0.0001
SM status			
Negative		Referent	
Positive	2.4	1.24–4.74	0.009
Unknown	1.9	0.41–8.50	0.4
LN status			
Negative		Referent	
Positive	3.4	1.59–7.17	0.001
Unknown	0.7	0.36–1.44	0.3

HR—hazard ratio, CI—confidence interval, PSA—prostate specific antigen, ISUP GG—International Society of Urological Pathology Grade Groups, SM—surgical margins, LN—lymph nodes.

## Data Availability

The data presented in this study are available within the article.

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
