# Peer review of "Effect of Clinical Parameters on Risk of Death from Cancer after Radical Prostatectomy in Men with Localized and Locally Advanced Prostate Cancer"

_cancers, 2022, doi:10.3390/cancers14082032_

Round 1
Reviewer 1 Report
In their manuscript, the authors performed univariate and multivariate analyses to better assess predictors of prostate cancer (PCa) cancer-specific mortality (CSM). An interesting retrospective work has been done on 2421 men who underwent radical prostatectomy for localised PCa.
From the analysis, the strongest predictors of CSM were ISUP group 5, lymph node invasion, stage pT3b-4, and age. The highest risk for death from PCa was discovered, from the multivaruate analysis, in the group ISUP 4 + ≥65y + pT3b-4, ISUP 5 + <65y + pT3b-4, ISUP 5 + ≥65y + pT3b-4. In accordance with the authors, this study presenting the stratification of PCa patients into subgroups according to GG, pathological stage, and age could help clinicians to identify the patient cohort with the most aggressive PCa who could benefit from more intensive treatment strategies. It represents an original paper of up-to-date interest. The manuscript is well structured and could be suitable for publication after some minor revisions in order to improve discussion on the need for the stratification of PCa patients into demarcated prognostic subgroups. In fact, prostate cancer is the major cause of death for men worldwide. Therapeutic options are represented by surgery, radiotherapy, hormonotherapy, and chemotherapy stand-alone or in a combination approach.
The challenge nowadays is to better understand the mechanism of cancer development, find real PCa markers, the best imaging exam, and develop new specific therapies in order to reduce the mortality of PCa (doi: 10.1016/j.urolonc.2020.03.007).
Better understanding predictors for CSM also help clinicians in better treatment in order to reduce overtreatment and its heavy side effects (doi: 10.1007/s11255-019-02107-3).
Author Response
Thank you for your positive comments on our manuscript. We made corrections in the discussion section (Line 267-270) and put in the reference list (29) one of your suggested publications. We hope that corrections will be acceptable for you.
Reviewer 2 Report
Thank you very much for giving me an opportunity for reviewing this manuscript. This manuscript assessed the long-term cause specific mortality (CSM) and other cause mortality (OCM) following radical prostatectomy in a total of 2421 patients with prostate cancer, which shows GGs, age, and pathological stage were the important factors for CSM. Combined assessment using these three factors easily stratified patients CSM and OCM. In favorable risk population (GG1, pT2a-pT3a, age<65), only few patients died with prostate cancer and other causes were the major reason of death whereas, in highest risk population (GG5 and pT3b-pT4), all patients died with prostate cancer. This manuscript is interesting for me and providing useful information to estimate long-term prognosis in patients undergoing radical prostatectomy. However, there are some points to be revised as following.
Major points:
Table 4 and Fig 1:
Table 4 and Fig 1 basically mean the same and Table 4 is busy for me. Authors could put Table 4 into supplementary materials and show only Fig1 in the main body of this manuscript.
Discussions
Age was one of independent factors for CSM. Authors need to comment how age was associated with CSM. Did younger patients received more intensive therapy for their prostate cancer?
Minor points:
Abstract
Line 14 and 22: “localized prostate cancer”.
This would be “localized and locally advanced prostate cancer” as shown in Line 76.
Materials and Methods
How did authors assess the survival analysis (5-year and 10-year mortality)? Kaplan-Meier methods? Authors need to specify this.
Table 1, Post RP treatment section: “Adjuvant, 63 (2.6)”
Is this adjuvant RT? Authors need to specify this.
Table 2, Parameter column
Could you add the number of patients in each parameter, like Age <65 (n= ----), ≥65 (n= -----). Also, I would recommend adding p-value for the comparison of curves.
Line 124: “pT3b-4 (HR3.1, 95% CI 1.32-7.38)”
A p-value is missing.
Line 128-129: Authors assessed the CSM using the combination of GGs, pT stage, and age.
In multivariate analysis, SM status was also an independent factor for CSM. Authors need to add comment why SM status was omitted in this analysis.
Author Response
Thank you for reviewing our manuscript. We have made corrections (in red) according to your recommendations.
Major points:
Table 4 and Fig 1:
Table 4 and Fig 1 basically mean the same and Table 4 is busy for me. Authors could put Table 4 into supplementary materials and show only Fig1 in the main body of this manuscript.
We agree that Table 4 duplicates information from Figure 1 and put Table 4 in supplements as suppl. Table 1.
Discussions
Age was one of independent factors for CSM. Authors need to comment how age was associated with CSM. Did younger patients received more intensive therapy for their prostate cancer?
Younger men do not have more aggressive pathological cancer characteristics compared to older men in our cohort. More intensive salvage treatment in younger men could be one of the possible explanations. However, we do not have such data at this moment. Probably better response to received treatment in younger men could be more important than treatment intensiveness on disease progression and CSM. We put comments in the discussion section: Lines 277-281.
Minor points:
Abstract
Line 14 and 22: “localized prostate cancer”. This would be “localized and locally advanced prostate cancer” as shown in Line 76.
We made a changes in Line 14 and 22
Materials and Methods
How did authors assess the survival analysis (5-year and 10-year mortality)? Kaplan-Meier methods? Authors need to specify this.
We used the Fine and Gray competing-risk regression analysis to model clinical parameters and follow-up data. The 5-yr and 10-yr CSM and competing-causes mortality from the time of prostate cancer and other causes stratified by important baseline predictors are summarized in Tables 2 and Suppl. Table 1. (Lines 89-91)
Table 1, Post RP treatment section: “Adjuvant, 63 (2.6)” Is this adjuvant RT? Authors need to specify this.
RT, as part of adjuvant treatment, was missed. We made corrections in Table 1.
Table 2, Parameter column Could you add the number of patients in each parameter, like Age <65 (n= ----), ≥65 (n= -----).
We additionally added age groups: ≥65 vs <65 in Table1. The numbers of each parameter are mentioned in Table 1
Also, I would recommend adding a p-value for the comparison of curves.
In our analysis, mortality rates are presented. To analyze the difference between OSM and CSM in each group in our opinion do not present any specific information. On the other hand, such comparison required very specific statistical approach and we decided not to add a p-value analysis.
Line 124: “pT3b-4 (HR3.1, 95% CI 1.32-7.38)” A p-value is missing.
Data in Line 124 corrected
Line 128-129: Authors assessed the CSM using the combination of GGs, pT stage, and age.
In multivariate analysis, SM status was also an independent factor for CSM. Authors need to add comment why SM status was omitted in this analysis.
The significance of positive SM on CSM in men after RP is still controversial. Indeed, it is a more surgeon dependent factor in a majority of cases rather than cancer characteristics. For this reason, despite the significance of SM in multivariable analysis, we decided not to put this parameter into further analysis. (Lines 95-97)

Round 2
Reviewer 2 Report
Line 2-3: Title [Effect of clinical parameters on risk of death from cancer after radical prostatectomy in men with localised prostate cancer ]
This would be Effect of clinical parameters on risk of death from cancer after radical prostatectomy in men with localized and locally advanced prostate cancer
Line 250-254: Younger men do not have more aggressive pathological cancer characteristics comparing with older men in our cohort.
Does authors show this data in the Results section ? If not, could you show it somewhere in the Results.
Also "Younger men did not have----" would be correct.
Author Response
Dear reviewer,
thank you for your additional comments.
We have change title line, also we added additional information about cancer characteristics in age groups - results section and supplementary Table1 - as well as some redaction in discussion section according your recommendations.